# Transcriptome Analyses Identify Deregulated *MYC* in Early Onset Colorectal Cancer

**DOI:** 10.3390/biom12091223

**Published:** 2022-09-02

**Authors:** Olivia M. Marx, Marc M. Mankarious, Melanie A. Eshelman, Wei Ding, Walter A. Koltun, Gregory S. Yochum

**Affiliations:** 1Department of Biochemistry & Molecular Biology, Pennsylvania State University College of Medicine, Hershey, PA 17033, USA; 2Department of Surgery, Division of Colon & Rectal Surgery, Pennsylvania State University College of Medicine, Hershey, PA 17033, USA; 3Department of Pediatrics, Division of Hematology & Oncology, Pennsylvania State University College of Medicine, Hershey, PA 17033, USA

**Keywords:** early-onset colorectal cancer, RNA-sequencing, transcriptome, MYC, PVT1

## Abstract

Despite a global decrease in colorectal cancer (CRC) incidence, the prevalence of early-onset colorectal cancer (EOCRC), or those occurring in individuals before the age of 50, has steadily increased over the past several decades. When compared to later onset colorectal cancer (LOCRC) in individuals over 50, our understanding of the genetic and molecular underpinnings of EOCRCs is limited. Here, we conducted transcriptomic analyses of patient-matched normal colonic segments and tumors to identify gene expression programs involved in carcinogenesis. Amongst differentially expressed genes, we found increased expression of the *c-MYC* proto-oncogene (*MYC*) and its downstream targets in tumor samples. We identified tumors with high and low differential *MYC* expression and found patients with high-*MYC* tumors were older and overweight or obese. We also detected elevated expression of the *PVT1* long-non-coding RNA (lncRNA) in most tumors and found gains in copy number for both *MYC* and *PVT1* gene loci in 35% of tumors evaluated. Our transcriptome analyses indicate that EOCRC can be sub-classified into groups based on differential *MYC* expression and suggest that deregulated *MYC* contributes to CRCs that develop in younger patients.

## 1. Introduction

Cancers of the colon and rectum are estimated to rank third in terms of newly diagnosed cancers and are expected to be the third-leading cause of cancer-related deaths [1]. While the overall colorectal cancer (CRC) incidence has been in decline since the early 1990s, there has been a steady increase in the incidence of early-onset colorectal cancer (EOCRC) [2,3]. EOCRCs are primarily detected as a result of abdominal pain or rectal bleeding, as opposed to later-onset colorectal cancers (LOCRCs), which are commonly found through surveillance colonoscopy [4]. Though the majority of EOCRCs are diagnosed between the ages of 40–49 [5], the American Cancer Society only recently (2018) began recommending that CRC screenings begin at age 45 instead of 50 [6]. As such, at the time of diagnosis, EOCRC patients frequently present with later-stage tumors and often have metastatic disease [7].

While genetic predisposition in terms of family history and hereditary conditions are associated with approximately a third of EOCRC cases [2], our understanding of the genetic and molecular pathways that drive carcinogenesis in these patients is far from complete. Compared with LOCRC, EOCRCs are most frequently microsatellite stable, near diploid, and often have an increased mutational burden [2]. Furthermore, several studies indicate that there are differences in the frequency and underlying mutations in oncogenic drivers in EOCRCs versus LOCRCs [2,8,9,10,11,12]. Despite these distinctions, current treatments for EOCRC and LOCRC remain largely the same [13]. To identify biomarkers and new treatment strategies for EOCRC and to better understand why there continues to be an increase in incidence worldwide, more work is needed to define the molecular and genetic underpinnings of EOCRC.

Expression of the *c-MYC (MYC)* proto-oncogene is deregulated in over 50% of all cancers [14]. MYC is a basic-helix-loop-helix transcription factor that binds to MAX to control the expression of numerous genes involved in cell cycle progression, metabolism, cell survival, and ribosome biogenesis [15,16,17,18]. Overexpression of MYC has been shown to contribute to early carcinogenesis in cancers, including prostate [19] and gastric [20]. Furthermore, studies suggest that MYC overexpression can promote field cancerization, which is thought to be a cancer-initiating event in which tissue regions are altered to promote increased proliferation [21]. Mechanisms that contribute to *MYC* deregulation in cancers include retroviral insertions, chromosomal translocations, locus amplification, and hyper-activation of oncogenic signal transduction pathways [22,23]. In addition, recent studies implicated the long-non-coding RNA (lncRNA) *PVT1* as a regulator of oncogenic MYC [24,25,26]. *PVT1* is located approximately 65 kb downstream from *MYC* at the 8q24 chromosomal locus, and it is frequently co-amplified in those tumors displaying gains in *MYC* copy number [25]. Moreover, *PVT1* binds and stabilizes MYC which in turn activates *PVT1* gene expression in a positive feedback loop to promote carcinogenesis [24,25]. Though it is well documented in the literature that MYC is a driver of colorectal cancer [23,27], its potential role in EOCRCs is less clear.

Large publicly available cancer datasets such as The Cancer Genome Atlas provide valuable resources for genomic and transcriptomic sequences of cancers. However, for many cancers, these resources lack patient-matched, non-cancerous samples [28]. The inclusion of patient-matched normal tissues is necessary to control for demographic or tissue-specific factors that could contribute to altered gene expression patterns in tumors [28,29]. Here, we use genome-wide RNA-sequencing (RNA-seq) on surgically resected tumors and adjacent colonic segments in a cohort of patients with EOCRC. We uncover evidence that the expression of *MYC*, and its downstream targets, are upregulated in tumors and find that tumors can be sub-classified based on differential *MYC* expression. We also detected elevated expression of *PVT1* in most tumor samples and gains in the *MYC/PVT1* genomic locus in a third of the samples. We found that those patients who had high-*MYC* expressing tumors at the time of surgery were older in our cohort and were obese or overweight. For the first time, these results implicate *MYC* as an oncogene in EOCRC. 

## 2. Materials and Methods

### 2.1. Specimen Collection 

This study was approved by the Pennsylvania State University College of Medicine Institutional Review Board (IRB Protocol No. PRAMSHY98-057). Prior to surgery, patients gave informed consent to have surgically resected intestinal tissues and blood to be collected and stored in the Inflammatory Bowel and Colorectal Disease Biobank (IBCRD) that is maintained within the Department of Surgery at the Pennsylvania State University College of Medicine. Patients in the study were under the age of 50 years old at the time of surgery, which was performed at our institution between 2015–2021. A flowchart of patient selection is included in Appendix A. Blood was drawn prior to surgery, and resected tissues were collected from the gross pathology lab and transported to our lab for processing. Full-thickness sections of rectum, cecum, sigmoid colon, ascending colon, and descending colon were collected, and aliquots were frozen in liquid nitrogen, fixed in formalin, and stored in RNAlater (Thermo Fisher, Waltham, MA, USA) preservative or embedded in paraffin for immunofluorescence analysis. Microsatellite stability was assessed through immunohistochemical staining of tumor samples for nuclear expression of four mismatch repair (MMR) proteins MLH1, MSH2, MSH6, and PMS2. Tumors were characterized as microsatellite unstable if defects in expression of MMR proteins were detected. Optional genetic testing using a microarray platform was performed for patients with microsatellite instability to confirm and diagnose Lynch syndrome. 

### 2.2. Indirect Immunofluorescence

Formalin-fixed tissue samples were embedded in paraffin and cut into 5 µm thick slices and placed onto VWR superfrost plus slides. Slides were dewaxed with xylene (two 5 min washes) and rehydrated with ethanol and water-graded washes (100% ethanol for 3 min, twice; 95% ethanol for 3 min, twice; 70% ethanol, once; 50% ethanol, once; water 5 min). Antigen retrieval was performed in 10 mM citrate buffer (pH 6) in a steamer for 20 min. Cooled slides were washed with PBS and blocked for 1 h in 5% donkey serum. Slides were incubated with primary antibodies against c-MYC (1:200 dilution, Cell Signaling Technology antibody #D84C12) in 5% donkey serum overnight. Slides were then washed in PBS. Secondary antibodies (Alexa Fluor 568 donkey anti-rabbit, Invitrogen, Waltham, MA, USA, #A10042) and DAPI (ThermoFisher #62248) were incubated on slides for 1.5 h. Slides were washed with PBS and mounted with mounting buffer (Invitrogen #P36982). Confocal fluorescent images were obtained at 63× magnification. Cells from each slide section were scored from 0–3 based on nuclear MYC staining intensity, and each cell value was averaged for a final score. This analysis was performed in triplicate for each sample.

### 2.3. RNA-Sequencing 

Tumors and adjacent full-thickness colonic tissues (*n* = 21 pairs) were collected in an Eppendorf microcentrifuge tube containing RNAlater and homogenized with a pestle. Chloroform was added, and the samples were centrifuged at 12,000× *g* for 15 min. The aqueous layer was subjected to further purification using an RNAEasy Mini Kit (Qiagen, Germantown, MD, USA), and RNA integrity values were assessed using a Bioanalyzer (Agilent, Santa Clara, CA, USA). Samples were further selected for transcripts that were poly-adenylated, and cDNA libraries were sequenced on a MiSeq2500 (Illumina, San Diego, CA, USA) sequencer as described previously [30,31,32]. Paired-end reads were generated at a read length of 100 nucleotides. Read quality was assessed with the program FastQC (v11.8, Babraham Bioinformatics, Cambridge, UK), all Phred scores were above 30, and GC bias was not detected in the samples. An average of 32 million reads per sample were aligned to the human genome (build hg38) using STAR version 2.7.3 (Cold Spring Harbor Laboratory, Cold Spring Harbor, NY, USA) with default parameters [33]. Aligned reads were counted using HTSeq [34]. Samples contained an average of 88% uniquely mapped reads. Quality assessment of the RNA-seq libraries is presented in Appendix A. 

### 2.4. Hierarchical Clustering and Differential Gene Expression Analysis 

The R package DESeq2, version 1.32.0 (Boston, MA, USA) [35], was used to normalize gene expression values and to identify differential expression of genes between all tumor and all colonic control samples. Differentially expressed genes were hierarchically clustered and visualized with gene-scaling for each row using the pheatmap function in R. Gene set enrichment analysis (GSEA) for cancer hallmarks was performed with 1000 phenotype permutations on rlog normalized counts from DESeq2 output on all tumors compared to all controls [36]. Gene ontology (GO) categories were assessed in R using package EnsDb.Hsapiens.v86 to convert ENSEMBL gene IDs to ENTREZ ID or gene symbol. The clusterprofiler package was used to identify ontology groups, and Enrichplot was used to visualize ontology results in R [37,38]. In order to calculate the log2 fold-change of expression in each tumor versus the patient-matched control, the DESeq2 function rlog was first used to determine the log2 normalized expression values, and then gene expression values from the adjacent colonic tissue were subtracted from values obtained from the corresponding tumor.

### 2.5. Cell Type Deconvolution 

In order to estimate the proportions of cell types in each sample, cellular deconvolution was performed as previously described [31] using xCell [39]. Results were plotted in R using ggplot2 [40]. We calculated significance between cell proportion scores using a paired Wilcoxon test for tumor versus adjacent normal samples, and we used an unpaired Wilcoxon test to compare high and low-*MYC* expressing tumors.

### 2.6. Analysis of Transcriptomes in The Cancer Genome Atlas (TCGA) 

Transcriptome data were retrieved from the TCGA COAD (colorectal adenocarcinoma) study (TCGA Research Network. Available online: https://www.cancer.gov/tcga accessed on 1 October 2021) using TCGAbiolinks package in R [41]. A total of 521 HTSeq-Counts files were filtered to isolate 474 primary tumor samples. Age at diagnosis under 18,250 days (approximately 50 years), was considered EOCRC (*n* = 55), and age above 18,250 days was considered LOCRC (*n* = 419). Count data were normalized using DESeq2 [35] and EnsDb.Hsapiens.v86 was used to convert ENSEMBL gene IDs to gene symbols. 

### 2.7. Reverse Transcription and Quantitative PCR 

cDNA was synthesized from 400 ng of purified RNA from each patient sample using the Verso cDNA synthesis kit (Thermo Fisher, Waltham, MA, USA) with random hexamer primers according to the manufacturer’s instructions. cDNAs were diluted to 10 ng/µL and subjected to PCR amplification with SYBR green chemistry using reactions and parameters previously described [30]. The following oligonucleotide sequences were used in the reactions: *MYC*; forward, AACTTGACCCTCTTTGGCAGCA and reverse, GCAAACCTCCTCACAGCCCAC; *β-ACTIN*; forward, GAGCATCCCCCAAAGTTCACAATG and reverse TGGCTTTTAGGATGGCAAGGGACT. *MYC* values were normalized to *β-ACTIN*, and fold change in expression was calculated using 2^−ΔCt^. The resulting qPCR values were compared with the normalized *MYC* expression values from the RNA-Seq analysis, and Pearson correlation was used to assess agreement between the approaches.

### 2.8. Copy Number Variation Analysis 

Genomic DNA was isolated from 5–25 mg of flash-frozen tissues and tumors using the QIAamp DNA minikit (Qiagen, Germantown, TN, USA) according to the manufacturer’s instructions. Quantitative PCR was performed using 20 ng of genomic DNA as the template in 20 µL reactions using the QuantStudio3 Real Time PCR system (Applied Biosystems, Waltham, MA, USA). The *MYC* genomic locus was evaluated using the Taqman probe Hs02602824_cn (Thermo Fisher), and the *PVT1* genomic locus was evaluated using Taqman probe Hs06232749_cn. The *RNASEP* gene (Thermo Fisher, 4403326) served as an internal reference. Reactions were performed in quadruplicate, and non-template controls were included. Results were analyzed with CopyCaller (Applied Biosystems, version 2.1). 

### 2.9. Statistical Analyses 

Statistical analyses were performed using R software v4.1.1. Two-tailed statistical tests were determined to be significant at *p* < 0.05. Significance of clinical correlates were determined using Fisher’s Exact test. Pearson’s correlation was used to determine the relationship between gene expressions. Spearman’s correlation was used to determine the relationship between gene expression and copy number variation. Paired Wilcoxon tests were used to compare gene expression in patient-matched normal colonic segments and tumors. All methods were carried out in accordance with relevant guidelines and regulations. 

## 3. Results

### 3.1. Patient Cohort

To define genes with EOCRC-associated expression, we first identified a cohort of patients in our inflammatory bowel and colorectal disease biobank who were under the age of 50 and for which we had archived flash-frozen tumors and adjacent uninvolved full-thickness colonic segments. For simplicity, we will refer to the adjacent colonic samples as control segments. We found 21 such patients whose demographics are listed in Table 1. In summary, our cohort is predominantly (90%) Caucasian, 57% male, and 76% overweight or obese, and these patients did not receive chemotherapy prior to surgery. Five patients (24%) had additional intestinal diseases, with four patients (19%) diagnosed with diverticulitis and one (5%) with inflammatory bowel disease. Tumors were resected throughout the colon and rectum, although no tumors were available from the transverse colon. An immediate family history of colorectal cancer was reported in nine patients (43%). Four patients (19%) exhibited defects in mismatch repair proteins, two of whom were diagnosed with Lynch syndrome after further analyses. All cancers were adenocarcinomas, with the majority being stage III or IV at the time of resection. 

### 3.2. Tumors and Adjacent Colonic Segments Display Distinct Transcriptomic Profiles

We isolated poly-adenylated RNAs from tumor and control samples and conducted RNA-seq to identify gene signatures associated with EOCRC. Using hierarchical clustering and principal component analysis (PCA) of normalized gene expressions of the top 4000 most variable genes in all samples, we found that with respect to their transcriptomes, tumors and controls form distinct clusters, with few exceptions (Figure 1A,B). Patient numbers 1 and 4 were diagnosed with Lynch syndrome, and patients 9 and 12 were found to have MMR defects but were not diagnosed with Lynch syndrome. In addition, tumor transcriptomes did not cluster based on the region of the colon from which they were resected (Figure 1A). We next used DESeq2 to identify differentially expressed genes in tumors versus controls and found that 1521 were upregulated (log2FoldChange > 1.5, adjusted *p* < 0.05) and 1542 were downregulated (log2FoldChange < −1.5, *p*-adjusted < 0.05) (Figure 1C). Gene ontology (GO) analysis on genes whose expression was significantly upregulated (log2FoldChange > 1.5, *p*-adjusted < 0.05) in tumors identified processes involved with nuclear division, chromosomal maintenance, and signaling receptor activity, amongst others (Figure 1D, Appendix A). The EOCRC GO signature is characteristic of highly proliferative cells. Cell type deconvolution analysis was performed to predict cellular populations of normal and tumor cells, confirming that the largest proportion of cells analyzed were either epithelium or smooth muscle cells. This analysis also identified differences in immune cell populations in tumors compared with normal samples (Appendix A).

### 3.3. MYC and Its Downstream Target Genes Are Differentially Expressed in EOCRC

We next applied gene set enrichment analysis (GSEA) to identify genes, or pathways, associated with hallmarks of cancer that were differentially expressed in the tumors versus controls [36,42]. Surprisingly, the expression of genes comprising the Wnt/β-catenin signaling hallmark set [42] was not significantly enriched in tumors as deregulation of this pathway has been previously described for both EOCRC and LOCRC [43,44] (Appendix A). Instead, we found significant enrichment (*p* < 0.05) of three hallmarks of cancer: unfolded protein response, DNA damage repair (Appendix A), and MYC targets (Figure 2A, Appendix A). We decided to further investigate MYC, given its established role as a driver of colorectal carcinogenesis [23,27]. We found that in addition to the enrichment of its downstream targets, the *MYC* oncogene itself was significantly enriched in tumors, and its differential expression clearly distinguished tumor samples from controls (Figure 2B). Through indirect immunofluorescence (IF) analysis, we found that expression of MYC is increased in tumors versus adjacent controls (Figure 2C,D). These findings suggest that deregulated expression of *MYC*, and its downstream targets, are involved in early-onset colorectal cancer pathogenesis.

### 3.4. Differential MYC Target Gene Expression Clusters Patients into Two Groups

We assessed whether the expression of genes comprising the MYC target hallmark set could identify sub-populations of patients in our study. To address this possibility, we used DESeq2 to normalize the data and to quantify fold-change expression differences of MYC hallmark genes in each patient-matched tumor and control tissue. Upon hierarchical clustering of these values, we found two discrete sub-classes of patients. The first subclass, comprising thirteen patients (62%), displayed increased expression of *MYC* and its downstream targets (Figure 3A). We will refer to these patients and tumors from these patients as high-*MYC*. In the second subclass of eight patients (38%), expression of *MYC* and most of its downstream targets were either down-regulated or unchanged in patient-matched tumors versus control segments (Figure 3A). We will refer to these patients, and their corresponding tumors, as low-*MYC*. Notably, as we grouped patients based on differential expression of *MYC*, and its downstream targets, tumors in the low-*MYC* group may still exhibit comparable levels of *MYC* transcripts to those expressed in tumors of the higher *MYC* group. As an independent validation of *MYC* expression, we performed RT-qPCR on isolated tumor and control RNAs. Plotting of RT-qPCR results versus normalized RNA-seq read counts confirmed the identification of high and low tumor profiles with respect to *MYC* expression (Figure 3B). Transcriptome data downloaded from The Cancer Genome Atlas (TGCA) of tumors from 55 colorectal adenocarcinoma (COAD) patients under 50 years of age show similar clustering into groups based on *MYC* hallmark signature gene expression, whereas in the over 50 age group, more samples have an intermediate expression of *MYC* targets (Appendix A). 

When considering our patient population as a whole, *MYC* expression is increased in all but two patient tumors versus matched control segments (Figure 3C). We used principal component analysis (PCA) of the top 5000 most variably expressed genes in tumors and found that those comprising the high-*MYC* sub-type were largely segregated from those in the low-*MYC* group (Figure 3D). When comparing high and low *MYC* tumors, we found that *MYC* is significantly upregulated in high-*MYC* tumors compared with low-*MYC* tumors (Wilcoxon test, *p* = 9.8 × 10^−6^, Appendix A). To further differentiate high and low-*MYC* tumors, cell deconvolution and GO analysis were performed on significantly (*p* < 0.05) deregulated genes in the high-*MYC* tumors compared with the low-*MYC* tumors. Notably, immune response gene signatures were downregulated in high-*MYC* compared with low-*MYC* tumors (Appendix A). In addition, we found that the expression of direct *MYC* target genes, including those involved in the polyamine synthesis pathway, were upregulated in the high- versus low-*MYC* tumors (*p* < 0.05, Appendix A). Importantly, when considering several clinical correlates, patients in our cohort with high-*MYC* tumors tended to be older and overweight or obese (Table 2). 

### 3.5. MYC and PVT1 Copy Number Alterations Are Found in Some Patient Tumors

Using RNA-seq data alone, it is difficult to assess which signal transduction pathways may impinge upon *MYC* to induce its expression. Because we found no associations in our samples with deregulated Wnt/β-catenin signaling, we sought to investigate other mechanisms which may contribute to increased *MYC* in the tumor samples. The *MYC* gene locus is subjected to copy number gains in numerous cancers, including CRC [45]. We, therefore, purified genomic DNA from matched tumors and controls from 17 patients and assessed the status of the *MYC* gene locus using a specific Taqman probe in quantitative PCR reactions. A probe against the *RNASEP* gene was included as an internal reference. We found that tumors from six patients (35%) displayed *MYC* copy number gains, with *MYC* copy numbers of 2.5 or higher (Figure 4A). Importantly, each of these six was among the high-*MYC-*expressing tumor population in our study. Tumors with copy number gains had increased levels of lymphovascular invasion, and they presented in older patients within our cohort (Appendix A). Notably, several other tumors were found to have gained in the 2.2–2.5 range, while all normal samples were within the 1.8–2.2 range. Six of the twelve tumors tested in the high-*MYC* sub-type did not contain greater than 2.5 copies of *MYC,* whereas none of the low-*MYC* tumors exhibited changes in copy number relative to uninvolved tissue (Figure 4B). 

To further investigate amplification in the 8q24 chromosomal region, we interrogated the *PVT1* locus in genomic DNA isolated from tumors displaying *MYC* copy number gains and their corresponding controls. We found that the *PVT1* locus was co-amplified in each of these tumor samples (Figure 4C). Together, these results suggest that *MYC* and *PVT1* copy number alterations account for increased *MYC* expression in half of the high-*MYC* tumors in our patient population.

### 3.6. Expression of PVT1 Correlates with MYC Expression

*PVT1* has been shown to potentiate oncogenic MYC function in CRC [24,25,26]. Since we have shown that *MYC* and *PVT1* are co-amplified in some tumors, we next evaluated *PVT1* expression in our RNA-seq dataset. We found that *PVT1* expression was increased in most tumors (90%) relative to adjacent control tissue (Figure 5A). Moreover, for most samples, *PVT1* and *MYC* gene expressions correlated (Figure 5B). In addition, we noted a correlation between differential expression of both *PVT1* and *MYC* in tumors versus control tissues in patient-matched samples (Figure 5C). These results suggest that regardless of copy number status, *PVT1* may be a positive effector of oncogenic *MYC* in early-onset colorectal carcinogenesis.

## 4. Discussion

In this study, we conducted RNA-seq analysis of patient-matched tumors and controls to identify molecular signatures associated with EOCRC. Hierarchical clustering of transcriptomes demonstrated that the vast majority of tumors clustered separately from control samples. That most tumors cluster separately from uninvolved tissues is in line with a previous study that reported transcriptomic profiles in organoid cultures derived from colonic mucosa and EOCRC [43]. Similarly, an earlier study reported that EOCRC gene expression profiles clustered separately from healthy controls [46]. Together, these results indicate that colonic gene expression profiles are significantly altered during the cellular transformation process to promote early-onset colorectal carcinogenesis. 

Mutations in components of the Wnt/β-catenin signaling pathway are prevalent in CRC, and aberrant Wnt/β-catenin signaling activates *MYC* expression [47]. As in LOCRC, mutations in the APC tumor suppressor have been reported in studies of EOCRC; however, such mutations were found in approximately 70% of tumors surveyed, which is lower than the 80–90% frequency reported in LOCRC [43,48]. Yan et al. reported deregulated expression of some Wnt/β-catenin targets in most tumor-derived EOCRC organoids in their study, which supports Wnt/β-catenin as a driver of early-onset tumors [43]. In contrast to this finding, our GSEA of the Wnt/β-catenin hallmark target gene set failed to identify the deregulation of Wnt hallmark genes in the tumor set as a whole (Appendix A). One difference between our study and that of Yan et al. is that we conducted transcriptomics on flash-frozen tumors and full-thickness control segments, whereas theirs was conducted on organoids, which are derived from the colonic epithelium [43]. It is possible that expression profiles obtained from surrounding stromal or immune cell infiltrate in our samples mask a Wnt/β-catenin target gene signature in epithelial cells. Indeed, cell deconvolution analysis predicts that epithelial cells contribute only part of the gene signature, with smooth muscle, immune, and stromal cells, among other cell types, contributing as well (Appendix A). While we did not assess the mutational status of *APC* or other Wnt-signaling pathway genes in our analysis, the lack of a deregulated Wnt/β-catenin target gene signature argues against this pathway as a significant driver of EOCRC in our surveyed population. 

Our GSEA identified *MYC* and its downstream targets as the top hallmark of cancer pathway that distinguishes early-onset versus control transcriptomes. To our knowledge, this is the first report to implicate increased *MYC* expression in EOCRC. Though somatic mutations in the *MYC* gene have been reported in EOCRC, whether these impacted *MYC* expression levels was not evaluated [48,49]. Moreover, there are conflicting reports on whether somatic *MYC* mutations are more common in earlier or later onset CRC [48,49]. Hierarchical clustering of our samples based on the differential of MYC hallmarks separated our patients into two clusters, higher and lower *MYC* expression. Future work will determine whether the somatic mutations in *MYC* influence its expression levels in the tumors of our cohort. 

We detected elevated expression of characterized MYC target genes in the high-*MYC* versus low-*MYC* tumors and found increased MYC protein expression via immunofluorescent staining of three patient’s tumors and adjacent normal tissues, arguing that MYC protein expression is also increased and that it is directly deregulating its downstream targets. Notably, we identified increases in MYC targets associated with polyamine synthesis, including ornithine decarboxylase 1 (*ODC1*), spermidine synthase (*SRM*), and adenosylmethione decarboxylase 1 (*AMD1*) [50] (Appendix A). Previous studies show aberrant MYC signaling is associated with increased polyamine synthesis, contributing to the deregulation of tumor metabolism and increased tumorigenesis [51,52]. When we investigated whether high-*MYC* tumors correlated with clinical parameters, we found patients with such tumors were in the overweight and obese categories (Table 2). Interestingly, the unfolded protein response (UPR), another hallmark of cancer that was enriched in our cohort’s tumors (Appendix A), may be caused by obesity-induced endoplasmic reticulum (ER) stress [53,54]. MYC may regulate the UPR by increasing ER stress and directly regulating the transcription of UPR regulators [55]. In turn, the UPR may regulate MYC directly at the transcript level or through the promotion of MYC translation [55]. Therefore, obesity may be contributing to MYC activity and tumorigenesis in EOCRC in part through the UPR. In addition, obesity has been shown to increase intestinal *MYC* expression through the Wnt/β-catenin pathway [56]. The importance of identifying a molecular link between obesity and CRC is highlighted by a recent study that showed that obesity is strongly associated with an increased risk of EOCRC [57]. It is likely that a combination of dietary factors, increased gut inflammation, and the microbiome all contribute to EOCRC in all patients [58]; however, more work is needed to evaluate the relationship between obesity and genetics within EOCRC.

Copy number alterations (CNA) are often found at oncogenic loci throughout the genome [59]. Indeed, CNA at the 8q24.21 locus, which contains the *MYC* gene, is common in many cancers, including CRC [25]. Pan et al. reported *MYC* CNA and increased *MYC* gene expression in colorectal cancers from younger (<60 years old) patients [60]. However, Lee et al. reported that *MYC* amplification was not associated with the age of CRC patients [61]. Here, we found gains in *MYC* copy number (>2.5 copies) in 35% of tumors surveyed compared to controls. Importantly, we observed a correlation between *MYC* copy number and increased *MYC* expression. Previous estimates of low *MYC* copy number amplification (>2 copies) range between 8–15% in CRC [61,62,63]. Therefore, in our EOCRC patient population, *MYC* CNA occurs more frequently than expected and is likely one mechanism that accounts for increased *MYC* expression within tumors. Due to our sample size, it is difficult to conclude whether *MYC* CNA is acting as a driver or passenger of carcinogenesis. No correlation was found between overall disease stage and *MYC* CNA, though we did find that patients with lymphovascular invasion were more likely to exhibit *MYC* CNA (*p* = 0.028, Appendix A). In addition, *MYC* CNA was found in older individuals (*p* = 0.0007), which may be supportive of amplification at the *MYC/PVT1* locus in tumor progression rather than initiation.

Prior work found that in 98% of cancers with MYC copy number increases, the amplified region includes the *PVT1* gene [25]. We found that all tumors with *MYC* copy number gains also displayed gains in *PVT1,* which supports the notion that the chromosomal 8q24 region harboring these genes is subjected to co-amplification in tumors. Other work has shown that *PVT1* cooperates with MYC to promote tumorigenesis [24,25,26]. In addition, a recent study has found that the *MYC/PVT1* locus is epigenetically regulated and may be subject to hypomethylation in CRC, which may also contribute to increased *MYC* expression [26]. In our study, we found that *PVT1* expression is increased in tumor samples and that it correlates with increased *MYC* expression. This finding suggests that while the *PVT1/MYC* region is not amplified in the majority of EOCRCs, *PVT1* may nonetheless serve as an important cooperating oncogene with *MYC* in tumors that develop in this patient population. The molecular mechanisms accounting for increased *MYC* and *PVT1* expression in most tumor samples in our study are currently unknown, and this line of work will be addressed in future studies. 

In this study, we were able to detect a strong association between *MYC* levels and tumorigenesis despite the area of the colon from which the tumor was resected, arguing that *MYC* may be a critical factor in tumorigenesis. The association of high-*MYC* tumors and obesity may provide the beginnings of a molecular explanation for the rising incidence of EOCRC in developed countries, which also display increasing rates of obesity [64]. Though diagnostic testing can detect EOCRC early on, the American Cancer Society only recently began recommending that CRC screenings begin at age 45 instead of 50 [6]. Furthermore, there is a lack of compliance with CRC testing, especially in younger patients [65], highlighting a crucial need for EOCRC biomarkers to improve upon less invasive early detection options. A previous study suggested that MYC is among a group of genes whose epigenetic signature may be used as an early biomarker for detecting colorectal cancer [66]. Our work may indicate that *MYC* profiling is useful for the detection of EOCRC as well as LOCRC. Furthermore, we have shown that high and low-*MYC* tumors have distinct transcriptomes and thus, likely have distinct mechanisms of disease pathogenesis. Therefore, measuring MYC and identifying high, and low-MYC tumors could inform treatment options. More work is needed to define the molecular mechanisms in which MYC is deregulated in EOCRC and how MYC may contribute to EOCRC progression.

## 5. Limitations

As is the case with the majority of EOCRC transcriptome studies to date, ours also contains a relatively small sample size. Our samples were from different tissues throughout the colon, and we included patients with MSI, Lynch, and IBD-associated cancers, which are associated with distinct CRC molecular subtypes [67,68]. Full-thickness tissues were homogenized for sequencing, and thus, we cannot differentiate the transcriptomic signatures of different cell types. Furthermore, as there is no well-established model for EOCRC, we were unable to confirm molecular mechanisms and focus only on correlations within the data. Finally, as our patient population is composed of largely Caucasians, we are unable to address racial and ethnic disparities in our study. 

## Figures and Tables

**Figure 1 biomolecules-12-01223-f001:**
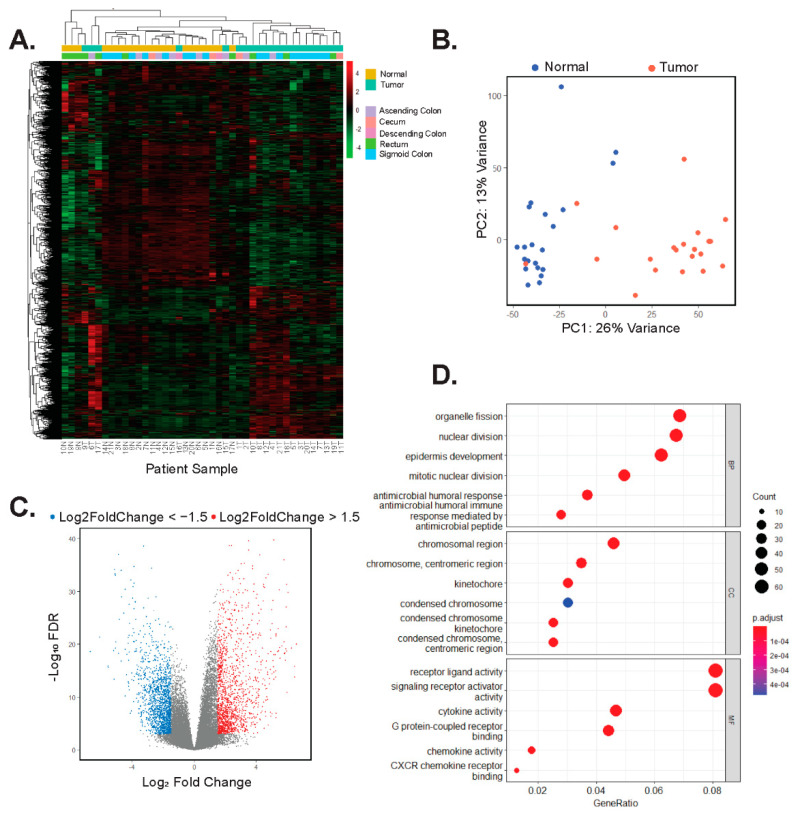
Tumors and adjacent colonic segments display distinct transcriptomic profiles. (**A**) Heatmap and hierarchical clustering of rlog normalized gene expression of the top 4000 most variable genes in tumors and adjacent control tissues. Patients are arbitrarily assigned 1–21 and N represents uninvolved tissues, whereas T represents tumor. (**B**) Principal component analysis (PCA) of the top 4000 most variable genes expressed in uninvolved (normal) and tumors. (**C**) Volcano plot of differentially expressed genes in tumors versus uninvolved control colonic tissues. Red points are 1521 genes whose expression is significantly upregulated (log2Foldchange > 1.5, *p*-adj < 0.05) and blue points are 1542 genes whose expression is significantly down-regulated (log2Foldchange < −1.5, *p*-adj < 0.05) in tumors versus controls. (**D**) GO analysis on the 1521 genes whose expression is upregulated in tumor samples. The top six results in each of the categories, biology processes (BP), cellular component (CC), and molecular function (MF) are shown in descending order.

**Figure 2 biomolecules-12-01223-f002:**
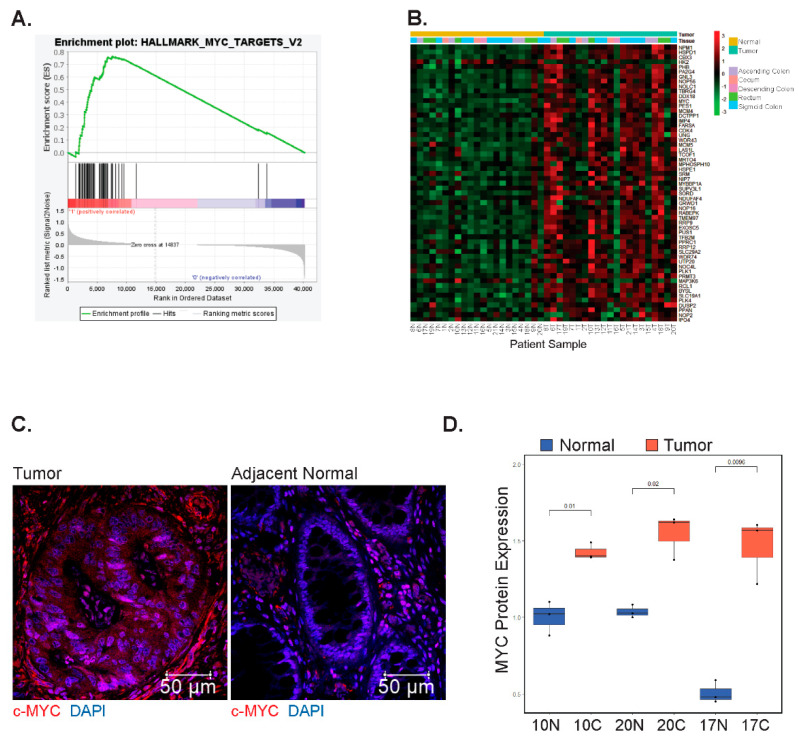
*MYC* and its downstream targets are differentially expressed in EOCRC. (**A**) Gene set enrichment analysis (GSEA) of genes comprising the hallmark MYC target set in tumors compared with patient-matched uninvolved colonic segments (Nominal *p*-value = 0.011, Normalized Enrichment Score: 1.52). (**B**) Heatmap of row-scaled gene expression of *MYC* and its downstream targets in tumors and adjacent colonic segments. Patients are arbitrarily assigned 1–21 and N represents uninvolved tissues, whereas T represents tumors. (**C**) Representative IF images of tumors and matched normal samples stained using a primary antibody against MYC. (**D**) Quantification of *MYC* positive nuclei from 3 fields of view per sample (N = 3 matched samples).

**Figure 3 biomolecules-12-01223-f003:**
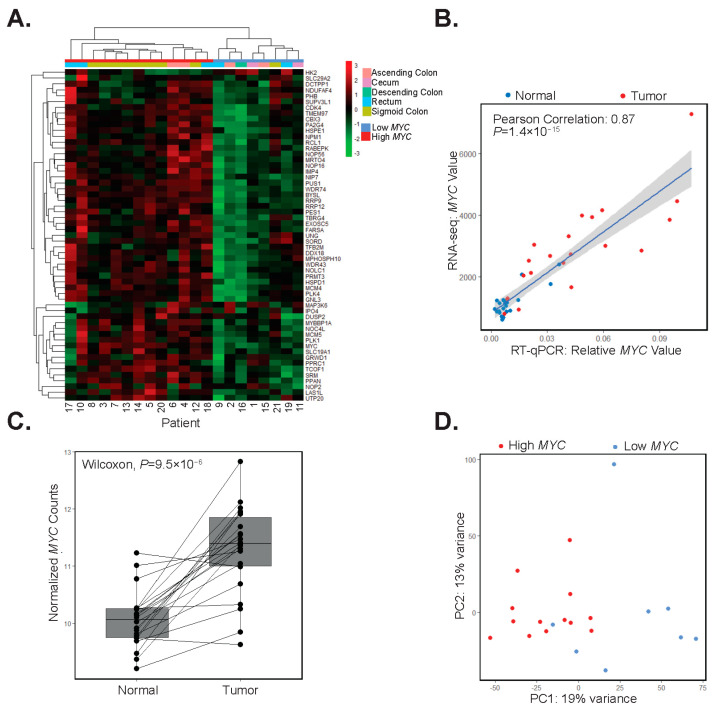
Differential MYC target gene expression clusters patients into two groups. (**A**) Hierarchical clustering of differential expression of *MYC* and its target genes in patient-matched tumors and adjacent colonic segments scaled by row. (**B**) RT-qPCR analysis of *MYC* and correlation with its expression in RNA-seq datasets in adjacent colonic segments (normal, blue points) and tumors (red points) (Pearson correlation 0.89, *p* = 1.4 × 10^−15^). (**C**) Boxplot of rlog normalized expression of *MYC* with lines connecting adjacent colonic segments (Normal) and tumor samples (paired Wilcoxon test, *p* = 9.5 × 10^−6^). (**D**) Principal component analysis of the top 5000 most variable genes in tumor samples with red points indicating high-*MYC* and blue points indicating low-*MYC* tumors.

**Figure 4 biomolecules-12-01223-f004:**
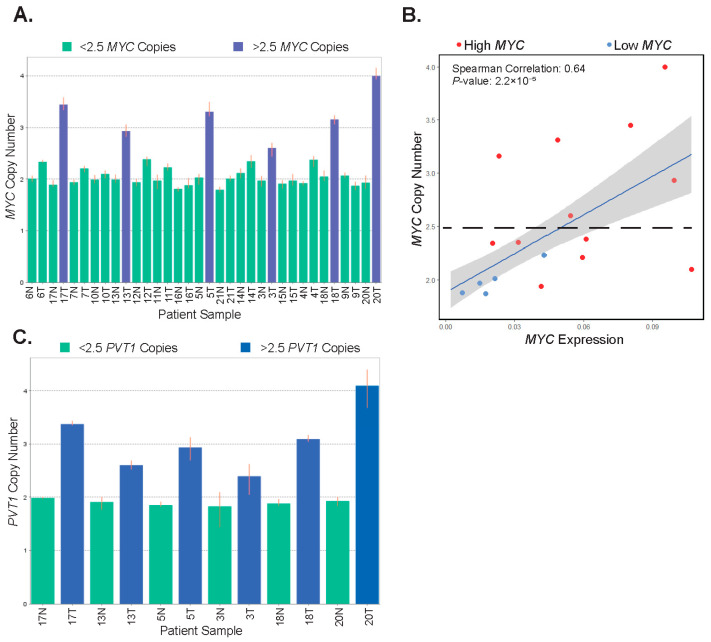
*MYC* and *PVT1* copy number alterations are found in some patient tumors. (**A**) Bar graph depicting the copy number of the *MYC* gene locus in patient-matched uninvolved colonic segments (normal, N) and tumors (T). Highlighted in blue are tumor samples with 2.5 or greater copy number. (**B**) Correlation of *MYC* copy number and *MYC* expression in tumors. Red points are tumors expressing high-*MYC* and blue points are tumors expressing low-*MYC* (Pearson correlation estimate 0.68, *p* = 5.45 × 10^−6^). (**C**) Bar graph depicting the copy number of the *PVT1* locus in the six patient-matched samples displaying gains in the *MYC* genomic locus in tumors (T). Uninvolved colonic tissues (normal, N) are shown as a reference. In (**A**,**C**), highlighted in blue are tumors samples with 2.5 or greater increases in copy number. Error bars represent the range of copy numbers detected in four technical replicates per sample.

**Figure 5 biomolecules-12-01223-f005:**
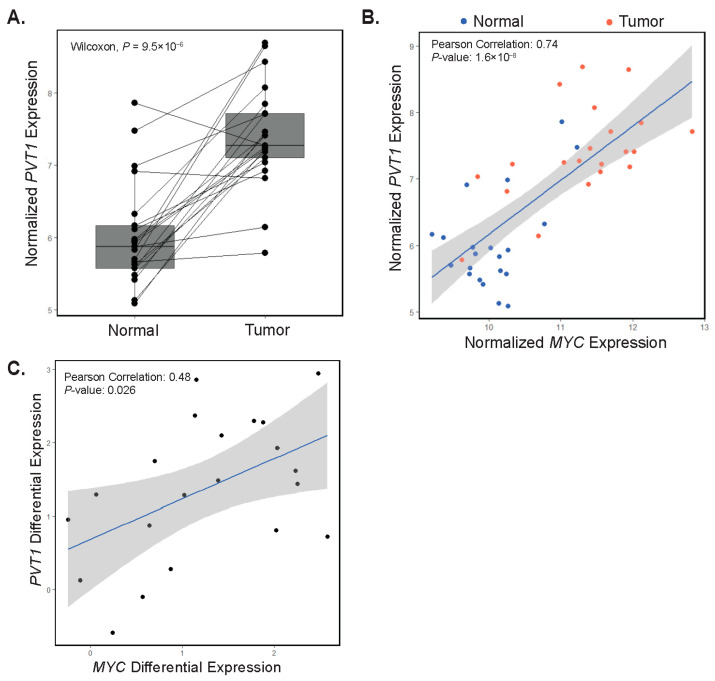
Expression of *PVT1* correlates with *MYC* expression. (**A**) rlog normalized expression plot of *PVT1* in patient-matched adjacent colonic tissues (Normal) and tumors (paired Wilcoxon test, *p* = 4.8 × 10^−6^). Each point represents a single patient with a line drawn between matched samples. (**B**) Correlation plot depicting normalized *MYC* versus normalized *PVT1* gene expression in adjacent colonic segments (Normal, blue points) and tumors (red points). (**C**) Correlation plot depicting *PVT1* and *MYC* differential expression in patient-matched adjacent colonic segments and tumors.

**Table 1 biomolecules-12-01223-t001:** Patient Demographics.

Parameter	N	(%)
Sex		
Male	12	(57)
Female	9	(43)
Race		
White	19	(90)
Black	1	(5)
Asian	1	(5)
BMI		
Underweight	0	(0)
Normal	5	(24)
Overweight	4	(19)
Obese	12	(57)
History of Diabetes	1	
History of smoking ^a^	8	(38)
Family history of colorectal cancer ^b^	9	(43)
Location		
Right Colon	6	(29)
Left Colon	1	(5)
Sigmoid	6	(29)
Rectosigmoid	2	(10)
Rectum	6	(29)
Stage		
I	6	(29)
II	3	(14)
III	10	(48)
IV	2	(10)
Adenocarcinoma	21	(100)
Signet/Mucinous histology	4	(19)
Lymphovascular invasion	7	(33)
Peri-neural invasion	6	(29)

^a^ Positive smoking status includes both current and former smokers at the time of surgery; ^b^ Positive family history for colon or rectal cancer in first degree family member.

**Table 2 biomolecules-12-01223-t002:** Patient demographics for low and high differential *MYC* expression.

Parameter	Low *MYC* Expression	High *MYC* Expression	*p*-Value
Patients, n	8	13	
Age at Diagnosis, median (IQR)	40 (39–44)	47 (46–48)	**0.043**
Sex			0.37
Male, n (%)	6 (75%)	6 (46%)	
Female, n (%)	2 (25%)	7 (54%)	
Race			0.13
White, n (%)	6 (75%)	13 (100%)	
Black, n (%)	1 (13%)	0 (0%)	
Asian, n (%)	1 (13%)	0 (0%)	
BMI			**0.047**
Normal or underweight, n (%)	4 (50%)	1 (8%)	
Overweight or Obese, n (%)	4 (50%)	12 (92%)	
History of Smoking ^a^, n (%)	4 (50%)	4 (31%)	0.65
Family History ^b^, n (%)	1 (13%)	8 (62%)	0.067
Located at Sigmoid or Rectum, n (%)	3 (38%)	11 (85%)	0.056
Overall Stage			0.53
I, n (%)	1 (13%)	5 (38%)	
II, n (%)	2 (25%)	1 (8%)	
III, n (%)	4 (50%)	6 (46%)	
IV, n (%)	1 (13%)	1 (8%)	
Signet/Mucinous component	4 (50%)	0 (0%)	**0.012**
Lymphovascular Invasion	3 (38%)	4 (31%)	1.00
Peri-neural Invasion	3 (38%)	3 (23%)	0.63

^a^ Positive smoking status includes both current and former smokers at the time of surgery; ^b^ Positive family history for colon or rectal cancer in first degree family member. Bold values represent statistically significant differences between high-MYC and low-MYC groups (Fisher’s exact test, *p* < 0.05).

## Data Availability

Data from The Cancer Genome Atlas (TCGA) are publicly available and can be retrieved from the following link: GDC (cancer https://portal.gdc.cancer.gov/.gov accessed 1 October 2021). The dataset generated during the current study are available in the gene expression omnibus (GEO) repository (GSE196006). Requests for materials should be addressed to Gregory S. Yochum.

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
