# Peer review of "Transcriptome Analyses Identify Deregulated MYC in Early Onset Colorectal Cancer"

_biomolecules, 2022, doi:10.3390/biom12091223_

Round 1

Reviewer 1 Report

In this manuscript, Yochum and collaborators perform expression and copy number analyses in a cohort of early-onset colorectal cancer patients with matched non-tumor controls.

Based on the differential expression analysis, they identify that (i) Myc is overexpressed in EOCRCs and that (ii) within the tumor cohort, there are Myc-high and Myc low tumors. They also report a lack of activation of WNT signaling.

The study is well structured, and the identification of Myc as a potential marker/driver of EOCRCs is of interest. Yet, in its present form, the manuscript is rather phenomenological and leaves unaddressed many important issues: what drives Myc over-expression in OECRCs? Is Myc the main driver, or are there also other recurrent mutations in these tumors? Is Myc level the only factor that discriminates Myc-high versus Myc-low tumors? Are other mutations/CNA recurrently/preferentially occurring in the Myc high tumors? Is the WNT pathway mutated in EOCRCs?

Minor points:

Figure 2B: please plot Myc levels separately and perform a statistical TEST to show that Myc is higher in tumors. The same analysis should be conducted when comparing Myc high tumors versus Myc-low tumors.

Figure 4B: there is one patient that belongs to the Myc-low group that shows Myc expression levels higher than four Myc-high patients. This does not seem to make sense. I would kindly ask the Authors to explain this.

Reviewer 2 Report

General comments/suggestions:

The authors show how MYC is perturbed and likely functionally important in EOCRC, however the authors failed to show if changes in MYC are also found in LORC patients ? Should the molecular alterations and factors described in this manuscript associated with MYC changes is solely occurring in EOCRC and not in LOCRC, then indeed this is an important finding. This needs to be clarified further and authors may consider to carry out this comparison using LOCRC vs EOCRC MYC analysis using TCGA data or qPCR analysis for LOCRC patients obtained from their own center ?

Similar to the comment above, the authors describe copy number as a potential mechanisms contributing to the increase level of MYC. However, it would be important to demonstrate either using TCGA or perhaps samples from their own center if indeed CNV associated with MYC is an event observed primarily/exclusively in the EOCRC cohort or is this also seen in LORC as well ?

The authors should perhaps tease out in the discussion section if CNV in MYC and its downstream effects could be an early event or perhaps a passenger event which happens due to environmental stressors and other factors such as obesity as a person ages ?

I would advise caution in making broad statements such as MYC profiling can be used to detect EOCRC as this MYC-associated change could also be observed in LOCRC?

Introduction:

- The authors here describe the importance of EOCRC within the current clinical context of CRC. This is then followed by describing the c-MYC proto-oncogene. It would be useful if the authors somewhat demonstrate a link between MYC impacts with early events in carcinogenesis OR even perhaps an association with early stage occurrence of other cancer types. This will ultimately present a strong rationale as to why further investigation in this is required.  

- Given the generic and very well understood role of transcriptome analysis in understanding molecular alterations underpinning disease aetiology, the need to explain this further in the intro is unnecessary. The authors following description of c-MYC and its potential role in EOCRC, should move to describing what they did when completing the introduction section.  

Results:

In general across the results section, when describing something as significant or not significant the p-value for the statistical analysis performed should be mentioned throughout the section.

Line 192-193: It is unclear as to why the authors decided to focus on Wnt when as the authors state, mutations in this pathway are highly prevalent in LOCRC’s, however the authors are focussing on EOCRC’s in this case ?

Line 239-241: p-values of the significant clinical correlates should be mentioned in the text

Line 236-238: The authors mention about figure S4 here, however this figure focusses on specific MYC target genes. Was GO for differentially expressed genes between low and high MYC carried out ?

Discussion:

Line 399-400: Consider revising this statement, to make a more generalised yet meaningful conclusion suggesting further studies warranted to extrapolate the functional role of MYC in driving molecular events underpinning early development of CRC ?

Reviewer 3 Report

In this study, Marx discovered that the hallmarks of cancer including unfolded protein response, DNA damage repair and MYC targets are enriched in gene set enrichment analysis with the 21 paired tissues from IBCRD. Among these cases, 13 patients (62%) displayed increased expression of MYC/downstream targets and 6 patients (35%) displayed MYC copy number of 2.5 gains. In addition, PVT1 expression was increased and correlated the expression of MYC in most tumors. In general, this retrospective study used the NGS and bioinformatics method to identify the signature of MYC/PVT1 in early onset colorectal cancer. It is an important finding to understand the development of CRC in younger patients. However, it is well known that MYC signaling pathway is hyper-activated in most of CRC. Besides, the PVT1 lncRNA has been shown as a novel epigenetic enhancer of MYC and a promising risk stratification biomarker in colorectal cancer (Molecular Cancer (2020) 19:155). Thus, it should be emphasized the importance of MYC/PVT1 axis in early onset, not in later onset colorectal cancer. Furthermore, the sample size is relatively small, and the MYC/PVT1 regulated molecular mechanisms in early onset CRC formation need to be confirmed. I think it is not ready to be published in its current state. The following are my comment and concern regards to the manuscript.

1. The authors need to consider the implication of the MYC/PVT1 in early onset colorectal cancer. What the application and how use the findings in clinical? Is MYC/PVT1 as diagnostic, prognostic or therapeutic target for early onset colorectal cancer? Whether the conclusion of this study could be the guidance for clinical treatment?

2. How about the status of MYC/PVT1 in later onset colorectal cancer?

3. Flow chart of the study and cohort selection should be included.

4. The author writes about the failed identification deregulation of Wnt Hallmark genes in tumors. An article mentioned the role of Wnt pathway members in body fat distribution, obesity, and metabolic dysfunction (Molecular Metabolism 42 (2020) 101078). The high-level expression of MYC in obese patients in the data might have a connection with the adipose role of the Wnt pathway. Although the assumption of stromal and immune cells might be possible (322-324 lines) because of maintaining tissue-wide Wnt signaling homeostasis, the adipocyte-specific loss of b-catenin could happen from CD45-/CD31 (immune cell). Is there any evidence that can be shown to support the assumption from the author? Also maybe the expression profile from the TCGA database can be utilized to compare the Wnt pathway in other databases.

5. Meanwhile, they also use the GO pathway but did not convey the genome systematical function comprehensively which might be helpful to give the whole picture of other related genes in MYC downstream and PVT1 gene.

Reviewer 4 Report

In Biomolecules manuscript 1814831 Marx et al. describe MYC upregulation in early onset colorectal cancer. Though MYC as a prototypical oncogene has been shown to be involved in numerous cancer types, data regarding this specific subtype have been limited. The authors present transcriptome analysis of 21 tumors and respective normal tissue. As a hallmark they observed significant upregulation of MYC and its targets in most tumor samples. Furthermore, they could in some cases attribute MYC hyperactivation to gene amplification. The study is of interest and well written, and the data appear to be solid. Below I raise some points that should be addressed before publication.

Major points:

1. How do MYC protein levels of tumors relate to control samples? Western blot or immunohistochemistry of a few samples could strengthen the main conclusion.

2. Same would be interesting for WNT/beta catenin: although the pathway does not seem to be significantly activated according to bulk RNA-seq it still may be on protein level or in a subset of tumor cells.

Minor points:

3. The authors should list all significantly regulated gene sets in a Supplementary Table.

4. Please disclose the patient identifiers with mismatch repair deficiency/Lynch.

5. As PVT1 is coregulated with MYC, the authors should include the lncRNA CASC11 = MYMLR in their analysis since it is also located within the MYC region and has been implicated in its regulation as well as in cancer progression.

6. More details on library prep and RNA-sequencing procedure should be given (kit, strandedness, read length, paired etc.).

7. Did the authors try to roughly estimate the cancer cell content in tumor samples? Tumors with low percentages may constitute outliers that cluster closer to the normal control (e.g. 16T).

8. If the 21 patients are consecutively numbered, who are patients 22 and 23 in Fig. 3A?

9. The authors could discuss MYC transcript upregulation by other means than copy number in more detail (they already mention MYC mutation as one possibility), since several of the high MYC tumors showed normal numbers.

Round 2

Reviewer 3 Report

I'm delighted to accept it.

Reviewer 4 Report

The authors responded adequately to the reviewers´ comments, and significantly improved the manuscript. Thus, I recommend its publication.